# Catchment-Scale Integrated Surface Water-Groundwater Hydrologic Modelling Using Conceptual and Physically Based Models: A Model Comparison Study

**Mohammad Bizhanimanzar \***  , **Robert Leconte and Mathieu Nuth**

University of Sherbrooke Water Research Group (USWRG), Department of Civil and Building Engineering, Faculty of Engineering, University of Sherbrooke, Sherbrooke, QC J1K 2R1, Canada; Robert.Leconte@usherbrooke.ca (R.L.); Mathieu.Nuth@usherbrooke.ca (M.N.)

**\*** Correspondence: Mohammad.bizhanimanzar@usherbrooke.ca

**Abstract:** This paper presents a comparative analysis of the use of an externally linked (MOBIDIC-MODFLOW) and a physically based (MIKE SHE) surface water-groundwater model to capture the integrated hydrologic responses of the Thomas Brook catchment, in Canada. The main objective of the study is to investigate the effect of simplification in representation of the hydrological processes in MOBIDIC-MODFLOW on its simulation accuracy. To this aim, MOBIDIC and MODFLOW were coupled in order to sequentially exchange the groundwater recharge and baseflow discharges within each computation time step. Using identical sets of hydrogeological properties for the two models, the coefficients of the gravity and capillary reservoirs in MOBIDIC were calibrated so as to closely predict the hydrological budget of the catchment simulated with MIKE SHE. The simulated results show that the two models can closely replicate the observed water table responses at two monitoring wells. However, in very shallow water table locations, the instantaneous response of the water table was not precisely captured in MOBIDIC-MODFLOW. Additionally, the simplified conceptualization of the unsaturated flow in MOBIDIC-MODFLOW resulted in overestimated groundwater recharge during spring and underestimation during summer. Moreover, the computational efficiency of MOBIDIC-MODFLOW, as compared to MIKE SHE, along with less required input data, confirms its potential for regional scale groundwater-surface water interaction modelling applications.

**Keywords:** surface water-groundwater interaction; integrated hydrologic models; MOBIDIC-MODFLOW; MIKE SHE; Thomas Brook catchment; water table fluctuations; groundwater recharge; unsaturated-saturated zone interactions

## 1. Introduction

Conjunctive water resources management entails having comprehensive knowledge about hydrological processes occurring in surface and subsurface zones and their interactions at the watershed scale. Integrated surface water-groundwater hydrologic models (ISGHM) with the capability to output the fine resolution spatiotemporal hydrologic response of the watershed are the tool of choice for such applications. During the past decade, a growing number of ISGHM with different levels of complexity have been developed. These ISGHM can be classified according to the subsurface flow process conceptualization into three groups.

In the first group, the flow in the subsurface (unsaturated and saturated) zone is described by using a three-dimensional variably saturated Richards equation. ParFlow [1] and CATHY [2] are

two widely used examples of this modelling category. This modelling approach offers the most complete description of the flow process (both in surface and subsurface zones) but it is operationally challenging at the watershed scale due to an intensive data requirement and very fine spatial and temporal discretization of the system which makes it computationally expensive.

In the second group, unsaturated flow represented by a one-dimensional Richards' equation that is coupled to a two (horizontal) or three-dimensional saturated flow model [3]. MIKE SHE [4], PIHM [5], PAWS [6] are widely used models based on this approach. This modelling approach is computationally less expensive than that of the first group (due to the simplifications made in modelling the unsaturated flow) which makes it more suitable for watershed scale integrated hydrologic modelling.

The third group includes externally linked surface water-groundwater models, in which the conceptual saturated flow module of a hydrologic model is replaced with a groundwater model such as MODFLOW [7]. SWAT-MODFLOW [8],TOPNET-MODFLOW [9], and GSFLOW [10] are examples of such integration approach. This group has the advantage of being computationally the fastest and offers a remarkably smaller amount of input data which makes it suitable for watershed scale studies. This is due to the fact that the unsaturated flow scheme of these models is not based on the solution of the Richards equation and no iterative procedure for coupling of unsaturated and saturated flows is involved [3]. Whereas the mentioned externally coupled models share similar formulation of groundwater flow i.e., MODFLOW, their unsaturated flow scheme is different, resulting in dissimilarity in number of calibration parameters of the models. SWAT distinguishes the flow process between the root zone and unsaturated flow through different formulations. Depending on the number of soil layers, the percolation occurs between the soil layers and eventually from the last soil layer to the vadose zone (the distance between the bottom of lowest soil layer to the top of the aquifer). The groundwater recharge occurs from the moisture available in the vadose zone assuming it as a linear reservoir i.e., discharge is linearly related to the moisture storage in vadose zone as also considered in the dual pore reservoir approach of MOBIDIC. A similar model structure is designed in GSFLOW, except for the calculation of unsaturated flow process which is handled by 1D Richards equation available in unsaturated flow package of MODFLOW. Unlike in SWAT-MODFLOW and GSFLOW, MOBIDIC simulates the flow process within the root zone and the flow process within the unsaturated zone is not explicitly conceptualized. This results in fewer number of calibration parameters of MOBIDIC in comparison to SWAT and PRMS (hydrologic model of the GSFLOW) which is beneficial in regional scale modeling where detailed information on soil characteristics of the watershed is not available.

Whereas the externally linked models showed that such integration of groundwater and surface water models has the advantage of capturing the spatio-temporal distribution of the groundwater recharge [11] and baseflow discharge [12], their application in shallow water table regions needs to be investigated. One of the particular characteristics of shallow water table regions is a rapid and significant response of the water table to exerted stresses (precipitation) due to the dynamic behavior of the specific yield. As discussed in [3,13], the magnitude of the water table fluctuations in these regions can be much greater than what would be expected with the constant value of the specific yield assumed in externally linked integrated models. Considering the important role of the shallow water table regions with regard to runoff generation in the river valleys and lowland regions of the catchments, a comprehensive evaluation of the effect of making simplifications in such integrated modelling approach is essential.

The objective of this paper is, therefore, to test the applicability of the externally linked surface-subsurface models to the Thomas Brook catchment, located in Eastern Canada, with complex heterogeneous aquifer structure, and interaction between surface and subsurface zones. To this aim, the surface water-groundwater modelling of the catchment was carried out following the integration of the hydrologic model MOBIDIC [14] with the widely used groundwater flow model, MODFLOW [7]. Simulations done with MOBIDIC-MODFLOW were then compared with those of the physically-based MIKE SHE, which we used as a base-case and also with available measurements. Such comparative analysis will enable answering the following questions:

1.　How do the simplifications in externally linked surface water-groundwater models (such as MOBIDIC-MODFLOW in this paper) affect the accuracy of the predictions, especially in shallow water table zones?

2.　Can MOBIDIC-MODFLOW with a low number of model parameters and high computational efficiency be regarded as an alternative to physically based integrated hydrologic models such as MIKE SHE in watershed scale surface water-groundwater modelling?

The ability of MOBIDIC's gravity/capillary reservoir approach to capture the near surface soil moisture dynamics was investigated in semiarid and humid sites comparing it to SHAW [15], a one-dimensional Richards based solver [14]. The model simulations and comparisons were performed for a single computational grid and the close match between model simulations and observations showed that the conceptual dual porosity soil moisture approach implemented in MOBIDIC was able to capture the dynamics of the near surface soil moisture if properly parameterized. However, whether such conclusion can be drawn from watershed scale simulations in which the dynamics of the soil moisture in gravity and capillary reservoirs are also affected by the water table fluctuations has yet to be evaluated. This study aims to further evaluate the suitability of the dual reservoir approach, in this case coupled to a numerical groundwater model (MODFLOW), at the catchment scale, as an integrated surface-subsurface modelling structure.

## 2. Study Area

The Thomas Brook catchment is located in the Annapolis Valley of Nova Scotia, Canada (Figure 1) and is one of the major fruit production regions of Canada due to its soil fertility and climatic characteristics [16,17]. The Annapolis Valley is approximately 100 km long and 10–15 km wide and is located along the Bay of Fundy, in between the North Mountain and the South Mountain ranges [18].

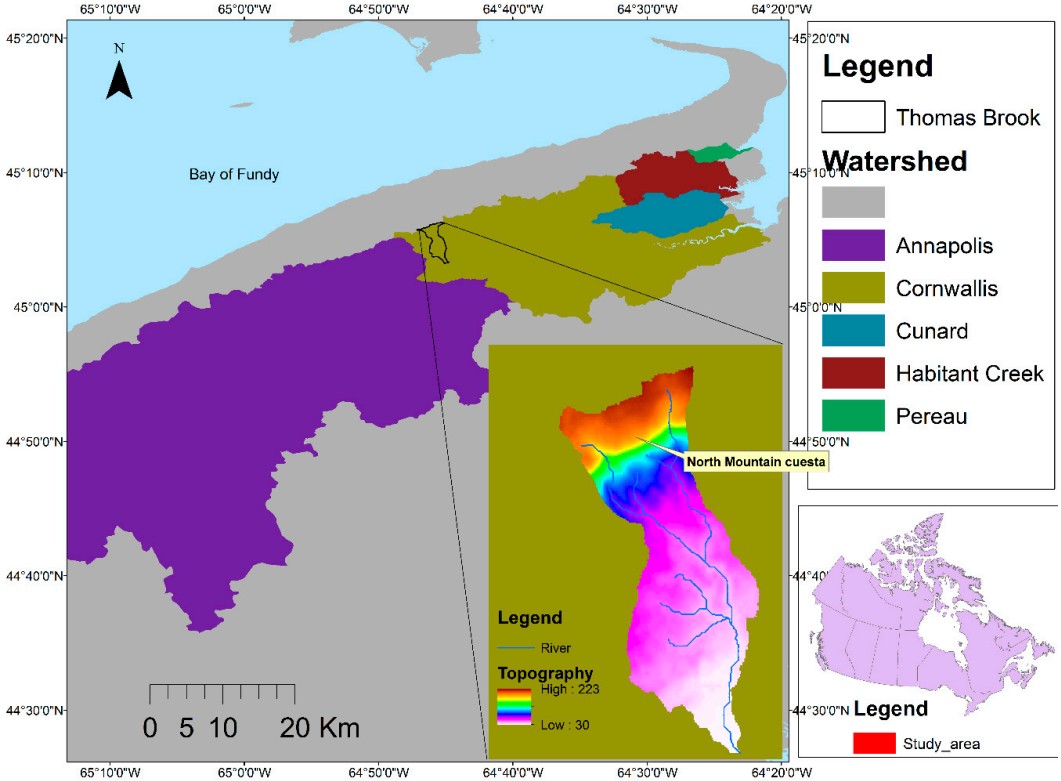

**Figure 1.** Location of Thomas Brook catchment in Annapolis Valley, Nova Scotia, Canada [17].

The Thomas Brook catchment is an 8 km$^2$ watershed located in the Cornwallis watershed (one of the five watersheds of Annapolis valley). The major part of land use is the farmlands (covering near

60% of the area), near 30% of the area is forested (in northern part of the catchment), and the remaining 10% is residential [18]. The elevation in Thomas Brook catchment ranges from 30 to 220 m over a length of about 5 km. The North Mountain cuesta with steep slope cuesta (elevation changes from 220 to 70 m over 1.5 km) is the reason of numerous existing springs at the foot of the slope which is the source of water for the residents [19]. The Thomas Brook river flows southwards and discharges into the Cornwallis river and eventually into the Bay of Fundy [19]. The groundwater flow also follows the topography from the North Mountain to the southern part of the catchment [19].

The 30-year average annual precipitation between 1971 to 2000 at the Kentville weather station, located about 20 km south-east of the catchment, is 1211 mm and the estimated potential evapotranspiration using Penman–Monteith method for the year 2005 is 689 mm [20]. About 78% of the total precipitation is rain which reflects the effect of the North and South Mountain ranges as well as the Bay of Fundy in making the Annapolis valley the warmest region of the Nova Scotia [20].

## 2.1. Aquifer Formations

Three types of aquifer formations are found in the Thomas Brook catchment: North Mountain, Blomidon, and Wolfville (Figure 2). The North Mountain formation overlies the Blomidon formation and is composed of a series of tholeiitic basalts characterized by numerous vertical fractures. However, due to the poor connection of the fractures, it represents the poorest aquifer formation of the catchment [19]. The Wolfville formation is composed of medium to coarse-grained sandstone, conglomerate, and siltstone. This formation is the most productive aquifer of the Thomas Brook catchment. The Blomidon formation is the second most productive aquifer in the catchment and has lithological properties that are similar to those of the Wolfville formation, but with more fine-grained layers, especially in the northern part near the North Mountain cuesta. All these formations are tilted with a 5° to 10° dip towards the Bay of Fundy [19].

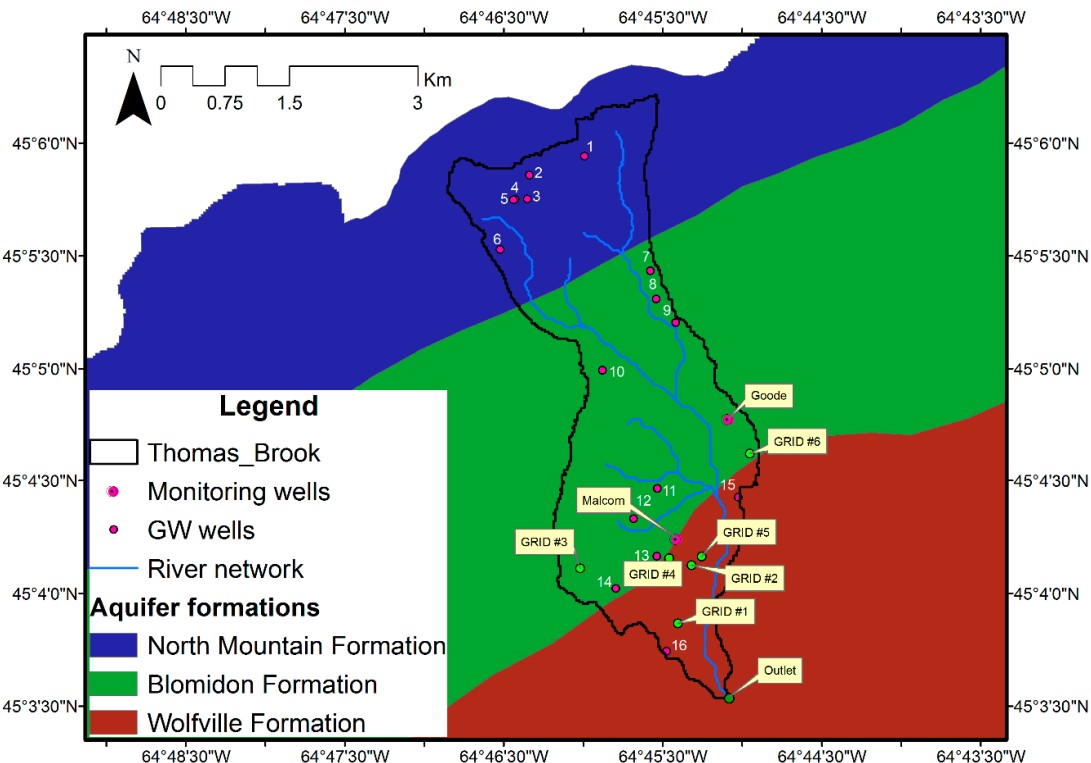

**Figure 2.** Aquifer formations along with the location of groundwater wells in the Thomas Brook catchment (modified after [17]).

## 2.2. Surficial Geology

Figure 3 depicts five geological formations representing the surficial geology of the watershed which is composed mostly of tills (covering near 70% of surface), glaciolacustrine formation (in the lower part of the catchment), colluvial deposits at the foot of the North Mountain formation, modern alluvium deposits, and glaciofluvial sands (underneath the glaciolacustrine formation) in the lower part of the catchment (Figure 3) [19]. The dominant surficial deposits (till and glaciolacustrine layers) have very low hydraulic conductivity (from $10^{-5}$ to $10^{-7}$ m/s) [21] which greatly reduces the magnitude of the groundwater recharge and also protects the underlying aquifer against contamination resulting from agricultural activities [19].

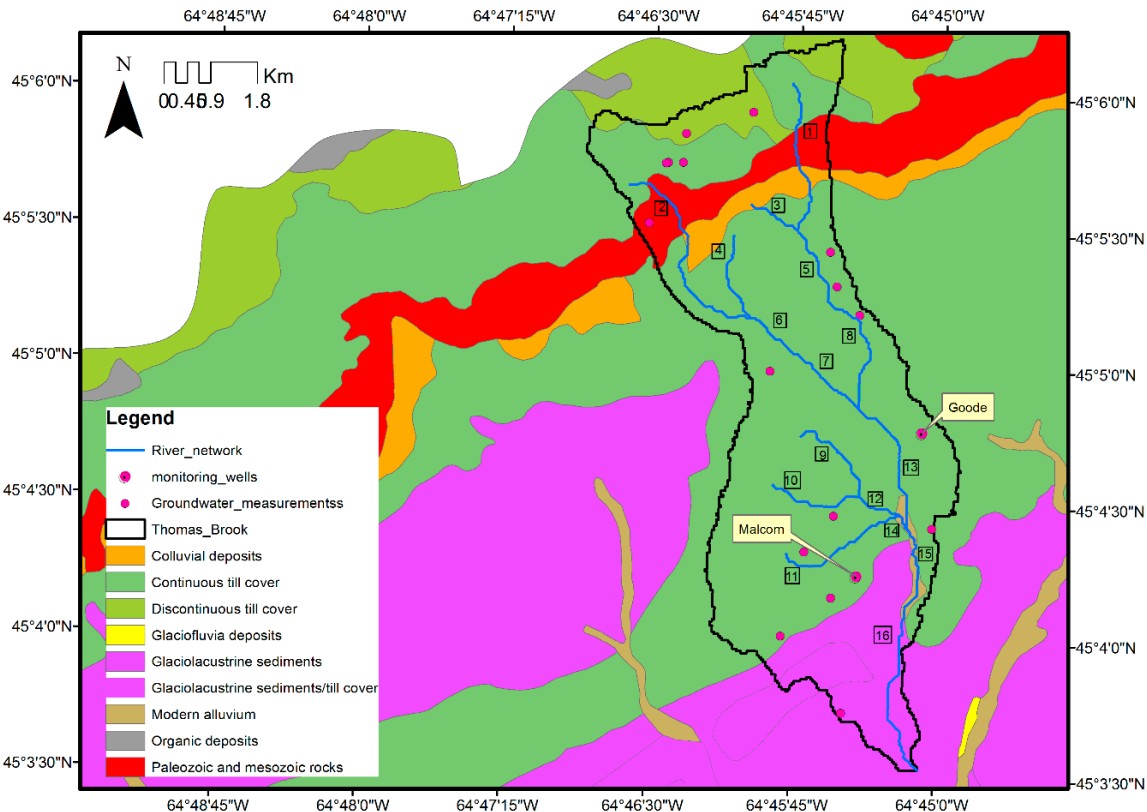

**Figure 3.** Surficial formations in the Thomas Brook catchment (modified after [17]). The number in rectangle box represents the reach numbers.

The detailed geological model of the catchment was given in [19]. They investigated the effect of increasing the representation of geological details (variable surficial layer thickness, aquifer hydraulic properties, and geological dip) on simulated streamflow discharge and groundwater recharge using the fully distributed physically-based CATHY hydrologic model. The simulation results revealed that the hydraulic conductivity of the aquifers has a notable impact on the predicted water table levels; however, the effect of adding other geological details such as the formation's dip or the aquifer porosity was minor. Additionally, remarkable improvements in the simulation results obtained by using the regional scale hydrogeological properties rather than the local scale field measurements were observed. The geological model of the catchment used in this study is depicted in Figure 4. The hydrogeological properties of the surficial and bedrock units are identical to those given in [19] (see Table 1). Whereas a detailed description of geological formations (thicknesses and saturated/unsaturated hydraulic properties) are required for integrated surface water-groundwater modelling with MIKE SHE, such information is not entirely needed in MOBIDIC-MODFLOW as it will be explained in the next section.

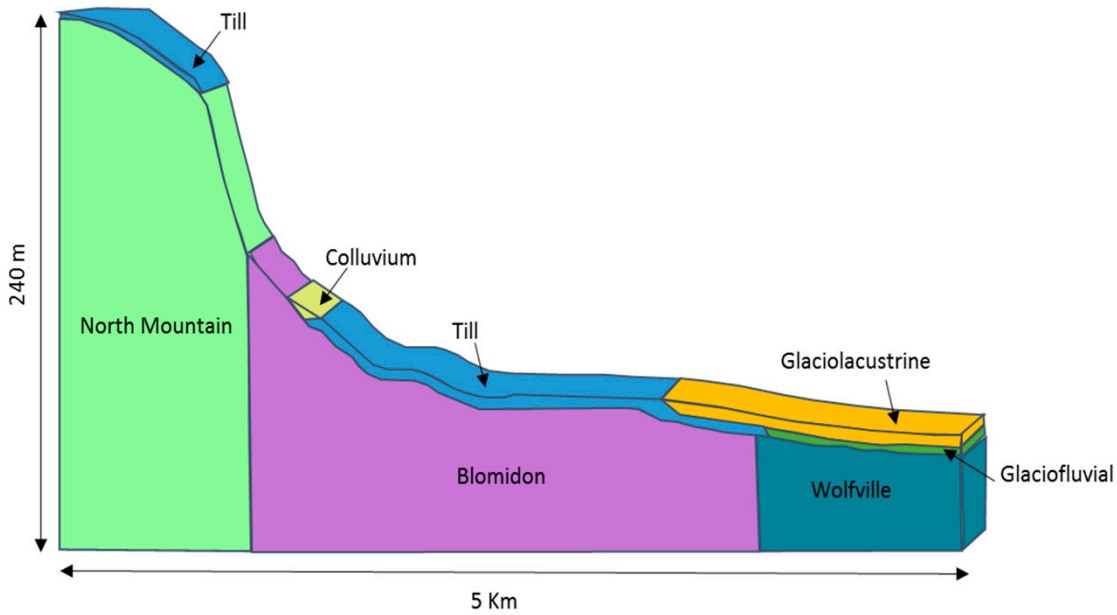

**Figure 4.** Schematic geologic model of the Thomas Brook catchment (modified after [19]).

**Table 1.** Hydrogeological properties of the formations in the Thomas Brook catchment [19].

| Formation | Saturated Conductivity (m/s) | Specific Storage (1/m) | Specific Yield in MIKE SHE ($\theta_{sat} - \theta_{fld}$) |
|---|---|---|---|
| North Mountain | $10^{-7}$ | $10^{-5}$ | 0.004 |
| Blomidon | $10^{-5}$ | $10^{-5}$ | 0.01 |
| Wolfville | $5 \times 10^{-5}$ | $10^{-4}$ | 0.03 |
| Tills | $10^{-7}$ | $10^{-3}$ | 0.02 |
| Colluviums | $10^{-5}$ | $5 \times 10^{-3}$ | 0.03 |
| Glaciofluvial | $10^{-5}$ | $10^{-2}$ | 0.03 |
| Glaciolacustrine | $10^{-7}$ | $10^{-3}$ | 0.013 |

## 3. Model Descriptions

The integrated surface water-groundwater modelling of the Thomas Brook catchment was carried out by using MIKE SHE, as the reference physically based model, and MOBIDIC-MODFLOW. The reference model was used to compare various outputs (recharge to groundwater, water table levels, infiltration, and evapotranspiration from unsaturated/saturated zones) of MOBIDIC-MODFLOW with the response of the catchment, as provided by MIKE SHE. In addition, for quantities where observations are available, such as streamflow discharges and groundwater heads at the two monitoring wells, the two models were compared with measurements.

### 3.1. MIKE SHE

MIKE SHE [4] is a widely used physically based distributed surface water-groundwater model capable of simulating the hydrological processes within the surface (overland and river flows) and subsurface zones [3]. The unsaturated zone in MIKE SHE is described by using the one-dimensional Richards equation which is coupled to the three-dimensional saturated flow [3]. The unsaturated and saturated zones are coupled by means of a water table correction procedure described in [3,22]. In this method, at each unsaturated flow time step, the simulated water table depth from the saturated module in previous time step is iteratively changed until the water balance of the profile falls below the prescribed threshold [3]. This is of great importance in modelling of water table fluctuations of shallow water table regions where the actual value of specific yield is smaller than its initial value [3]. A 2D diffusive wave and 1D kinematic wave approximation of Saint-Venant equations simulate overland flow and the flow in channels/rivers, respectively.

The modular structure of the model allows the use of different time steps for each module (seconds to minutes for the unsaturated flow and daily for the saturated flow) which facilitates the computational process. In the unsaturated zone and by using its own time step, the solution of the Richards equation updates the soil moisture profile between soil surface and the water table level simulated by the saturated flow module in the previous time period. By using the determined unsaturated zone moisture profile and the prescribed water table level; one can calculate the generated mass balance error of the soil profile. If the produced error falls below the user-specified tolerance, no water table adjustments are performed. Otherwise, the water table level is iteratively adjusted until the acceptable mass balance error in the soil profile is met. Once the iterative process is completed, the groundwater recharge is calculated and applied as the upper boundary condition of the saturated module. Then, the groundwater heads and the exchange flow between the aquifer and the river reaches are calculated and the simulation proceeds to the next time step.

The snowmelt process follows the degree-day method as the rate of melt linearly increases with rises in air temperature above a prescribed threshold (melting temperature) [23]. The evapotranspiration mechanism can take water from ponded water, the unsaturated zone, and directly from groundwater as detailed in [24]. The spatial discretization of the watershed into square grids allows for a grid-scale comparison of the hydrologic processes e.g., infiltration, recharge to groundwater, and evapotranspiration.

*3.2. MOBIDIC-MODFLOW*

3.2.1. MOBIDIC

Modello di Bilancio Idrologico DIstribuito e Continuo (MOBIDIC) is a distributed hydrologic model originally developed by [25] and recently enhanced by [14]. As is the case in MIKE SHE, in MOBIDIC, the watershed is discretized in square grids, as shown in Figure 5a. The hydrologic processes of each grid are calculated by using four inter-connected reservoirs i.e., canopy, surface water bodies (ponds, lakes), gravity, and capillary. The gravity and capillary reservoirs take into account the hydrological processes within the modeled soil layer i.e., infiltration, absorption, evapotranspiration, groundwater recharge, and capillary rise and also interact through moisture absorption [3]. The gravity reservoir holds moisture in the larger soil pores (exceeding the field capacity) and interacts with the underneath saturated zone through groundwater percolation [3]. The gravity reservoir can also discharge the available moisture to the adjacent downhill grid (interflow). The capillary reservoir corresponds to moisture in the finer pores (below the field capacity) and contributes to the evapotranspiration process. This conceptual formulation of the subsurface flow process results in a computationally more efficient model than one which is based on solving the Richards equation.

The computed overland runoff (as saturation excess and infiltration excess) of each computational grid is routed by using the steepest descent (D8) scheme (flow from a grid is transferred to one of its eight adjacent grids with the steepest slope) and eventually is propagated through the river network by using the linear reservoir approach.

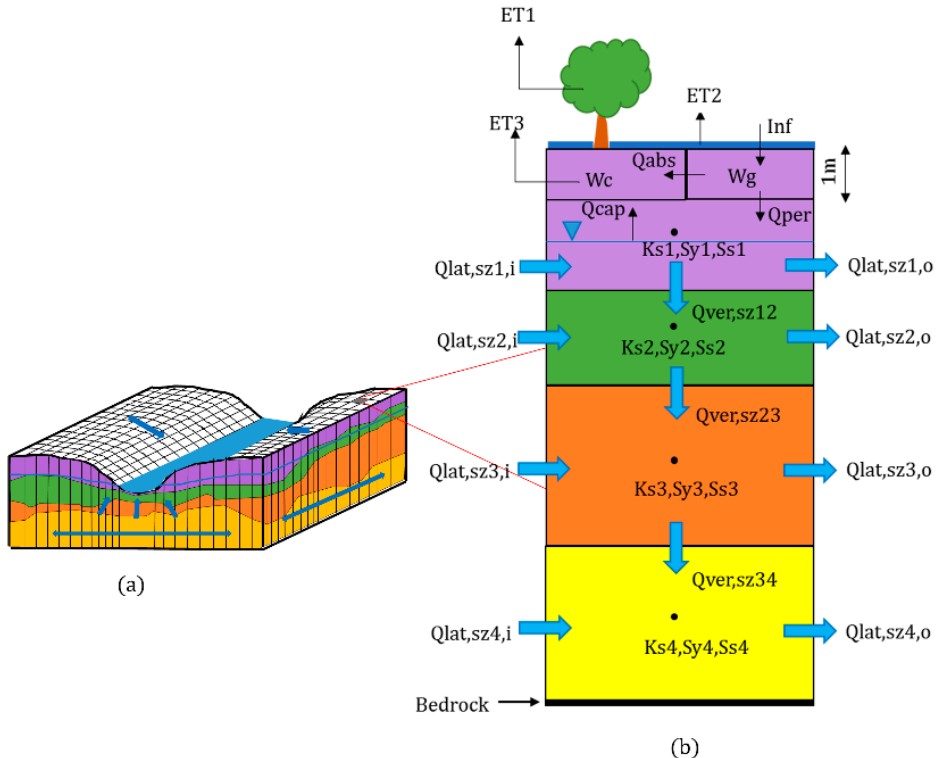

**Figure 5.** (**a**) Domain discretization in MOBIDIC-MODFLOW and in MIKE SHE; (**b**) schematic diagram of the hydrological fluxes in MOBIDIC-MODFLOW. Inf: Infiltration, ET1: Canopy transpiration, ET2: Surface water evaporation, ET3: Root transpiration, Wg: Gravity reservoir, Wc: Capillary reservoir, Qabs: Absoption flux, Qper: Percolation to groundwater, Qcap: Capillary rise, $Ks_i$, $Sy_i$, $Ss_i$: Saturated conductivity, specific yield, and specific storage of layer i, $Qlat,sz_i,i$: Lateral inflow to the layer i, $Qlat,sz_i,o$: Lateral outflow from layer i, $Qver,sz_{ij}$: Vertical flow from layer i to j. Note that the top 1 m of the soil profile is represented by MOBIDIC, while the underlying layers are modeled by MODFLOW.

### 3.2.2. MODFLOW

MODFLOW [7] is a widely used three-dimensional groundwater model. MODFLOW has a modular structure capable of simulating both flow and solute transport in complex confined and unconfined heterogeneous aquifers. The explicit formulation of boundary conditions such as the groundwater recharge, evapotranspiration, and flow to/from river beds makes it a suitable choice for modelling of hydrologic processes that interact with the saturated zone in surface/subsurface models such as SWAT-MODFLOW [12], GSFLOW [10], and MOBIDIC-MODFLOW which were used in this study.

### 3.2.3. Coupling MOBIDIC with MODFLOW

The flowchart describing the coupling procedure between MOBIDIC and MODFLOW-2000 is shown in Figure 6. At each time step, meteorological data (precipitation, temperature, wind speed, relative humidity, and solar radiation) are read and based on Penman–Monteith method, the potential evapotranspiration is calculated. The spatial distribution of groundwater recharge and groundwater heads, as well as the water level in river reaches obtained from the previous time step are passed on to MODFLOW and baseflow and groundwater heads are calculated. The overland flow module then routes the generated surface runoff from the grids and calculation of the hydrological process in MOBIDIC i.e., infiltration, absorption, evapotranspiration, recharge, and capillary rise commences. Then, the calculated groundwater recharge is transferred to MODFLOW and the calculated surface runoff discharge plus the baseflow discharge (calculated in MODFLOW) is transferred in the river routing module of MOBIDIC for further calculation of the streamflow discharge [3]. River stages are

calculated in MOBIDIC based on Manning's equation and on assuming uniform river water level for each river reach.

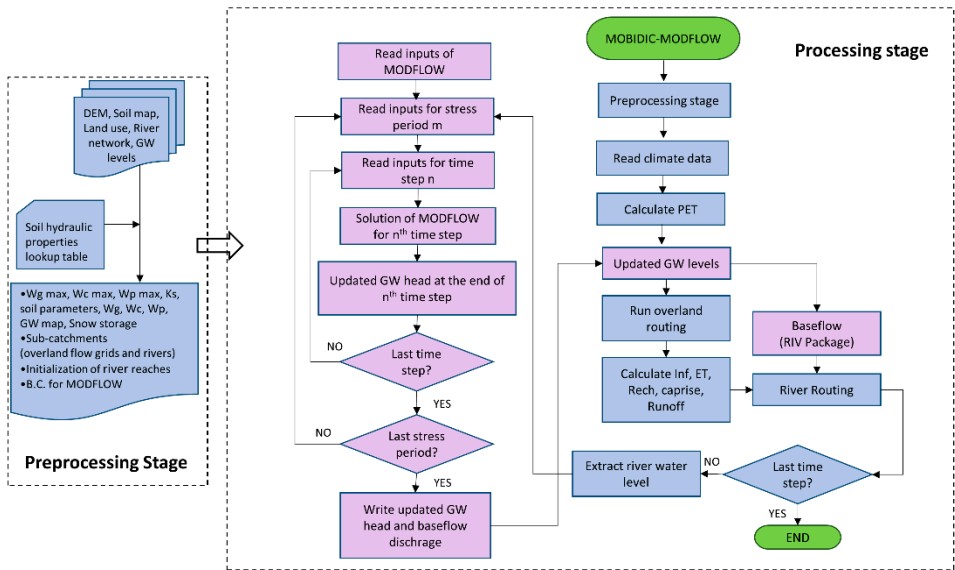

**Figure 6.** Flowchart showing integration procedure of MOBIDIC and MODFLOW. The blue and purple colors represent the process conducted in MOBIDIC and in MODFLOW, respectively.

MOBIDIC employs a uniform lateral discretization of the watershed domain, which results an identical planar dimension of the subsurface soil layers in MODFLOW. The vertical discretization of the computational layers, however, is based on geological layers. Therefore, the vertical soil profile of a computation grid in MOBIDIC-MODFLOW has two parts: (1) a near surface homogeneous soil layer for computation of hydrologic processes in MOBIDIC and (2) the geological layers for computation of saturated flow between layers in MODFLOW (see Figure 5b).

## 4. Modelling of the Thomas Brook Catchment

The integrated surface water-groundwater modelling of Thomas Brook followed the four-step procedure shown in Figure 7. The first step i.e., constructing the 3D geological model of the catchment is common to both MIKE SHE and MOBIDIC-MODFLOW. In this step, the topographical maps, as well as the stratigraphy of surficial and bedrock units, are imported into a geographical information system to create the 3D geological model of the catchment. The topographic map of the catchment was produced from Canadian Digital Elevation database at a resolution of 20 m (http://maps.canada.ca/czs/index-en.html), which is also the size of computational grid squares used in MIKE SHE and MOBIDIC-MODFLOW. The surface discretization of the model domain into 20 m square grids resulted in 28,835 computational grid squares in both models. The surficial and bedrock stratigraphy of the catchment was derived from [17].

The second step is the model set-up which involves processing the required inputs for the models. As shown in Figure 7, this step proceeds differently in MIKE SHE than in MOBIDIC-MODFLOW. In MOBIDIC-MODFLOW, the Digital Elevation Model (DEM) was processed to derive flow direction, flow accumulation, and the river network maps required for the surface water routing process. The derived river network was then used to identify the river cells in which an aquifer-river flow exchange may exist. In MIKE SHE, the river network is constructed in MIKE 11 which handles the river flow modelling aspect of the model. It is worth mentioning that in MIKE SHE and MOBIDIC-MODFLOW, the river-aquifer flow exchange is calculated based on the differences in river and groundwater level heads multiplied by riverbed conductance. However, in MIKE SHE, the river reaches are located on the edges of the two adjacent grids, which allow flow exchange between the river and grids on both

sides of the river. This is not the case with the MODFLOW's river package in which the exchange occurs only from river cells.

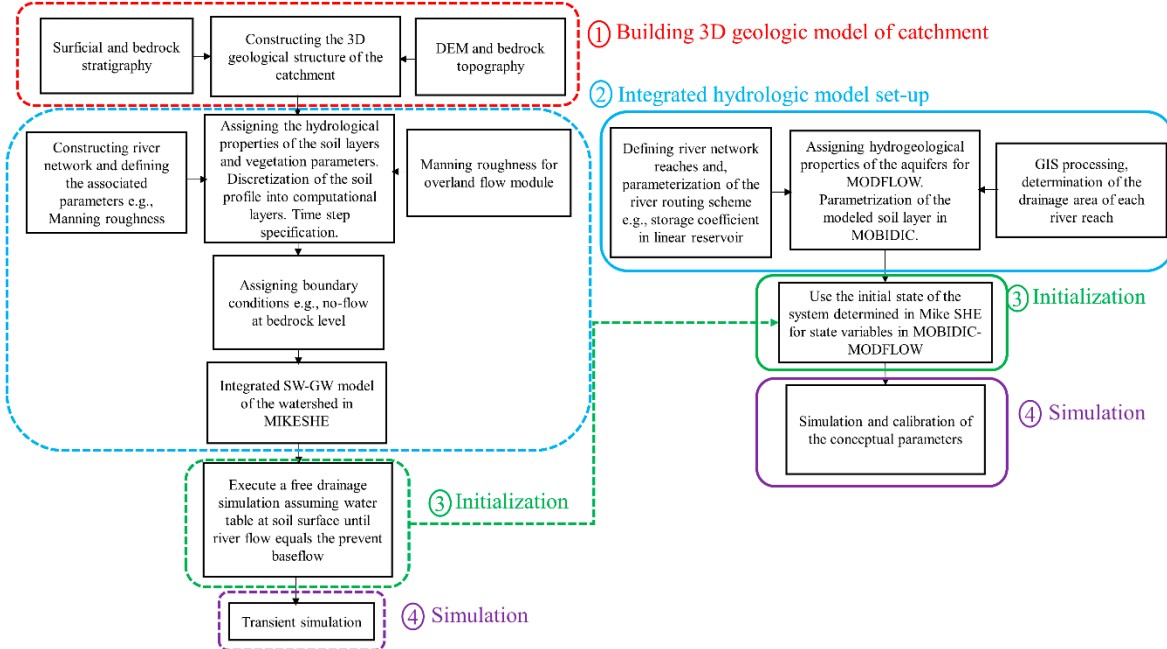

**Figure 7.** The step-by-step procedure of integrated surface water-groundwater modelling of the Thomas Brook catchment with MIKE SHE and MOBIDIC-MODFLOW. The left side (boxes in dash) represents the process followed in MIKE SHE and the right side (solid box) shows the corresponding steps in MOBIDIC-MODFLOW. Note that step 1 is similar in both models.

The numerical solution of the Richards equation in MIKE SHE requires the vertical discretization of the soil profile into computational layers. Layer thicknesses are usually thin near the soil surface to allow for an accurate representation of the infiltration and rainfall-runoff partitioning (especially in heavy rain periods) and is progressively coarsened with depth. In modelling the Thomas Brook watershed, the soil profile was vertically divided into four discretization zones, each with uniform node spacing. Zone number one (top 1 m of the soil) had a resolution of 1 cm. Zone number two (from 1 to 5 m below the soil surface) was discretized into 2 cm computational layers. Zone number 3 (from 5 to 15 m) had uniform layers of 5 cm thickness and zone 4 (from depth 15 m to the bedrock level) was discretized into 10 cm layers. Unlike the fine resolution of computation nodes in the unsaturated module of MIKE SHE, the computation of saturated flow was based on the assigned hydrogeological properties of each formation. Therefore, in a given time step, the water table level determines which computational nodes of the unsaturated zone are active (above the water table) and therefore used in the unsaturated flow module. Note that in MOBIDIC-MODFLOW, calculation of the hydrological processes and moisture dynamics of the gravity and capillary reservoirs (see Figure 5b) is performed over the top 1 m of the soil profile (root zone) and unlike Richards equation based approach in MIKE SHE, no vertical discretization of the soil profile is performed (see Figure 5b). The soil layer thickness is an input of the model and can be spatially non-uniform (grids can have different soil layer thickness). Having a wider or thinner soil layer would not change the generation of Horton's runoff in the model since this runoff mechanism is generated when the precipitation rate exceeds the saturated hydraulic conductivity [14]. However, generation of Dunne runoff is based on complete saturation of the soil profile and therefore can be affected by the layer thickness [3]. Additionally, since the reservoirs are not vertically distributed, the root water uptake is extracted from the entire capillary reservoir (one can envision it as a uniform water uptake by plant roots from the top 1 m of the soil). Therefore, the thickness of the soil layer can be defined as the maximum rooting depth of the computational grid's

vegetation type and its moisture state throughout the simulation period (and consequently generation of Dunne runoff) is regulated by the coefficients of the gravity and capillary reservoirs.

An important part of the model set-up step is parametrizing the evapotranspiration process in the models. Calculation of evapotranspiration process from the unsaturated zone in MIKE SHE follows the Kristensen approach [26], in which the temporal variation of vegetation characteristics e.g., root depth has to be known. Such data, however, were not available for the Thomas Brook catchment and a time-invariant root depth of 1 m was assigned for the simulation period. This decision was taken to make the temporal evolution of the root transpiration of the two models comparable. It should be noted that the magnitude of the evapotranspiration in MOBIDIC-MODFLOW is controlled by the coefficient of $\xi$ (see Table 2) which is a conceptual parameter in MOBIDIC. Additionally, similar to [19], it was assumed that the overland flow routing parameters (Manning roughness, detention storage) used in MIKE SHE were uniform for the entire catchment.

**Table 2.** List of input parameters for MOBIDIC-MODFLOW and MIKE SHE. The detailed definition of the parameters for MOBIDIC-MODFLOW and MIKE SHE are given in [14,24], respectively.

| Model Component | MOBIDIC-MODFLOW | MIKE SHE |
|---|---|---|
| Unsaturated flow | $-K_s$, $\theta_{sat}$, $\theta_{fld}$, $\theta_{res}$, $\psi_1$, $m$ for modeled soil layer <br> $-d[L]$: thickness of the modeled soil layer <br> $-\gamma[1/T]$: percolation to groundwater coefficient <br> $-\kappa[1/T]$: moisture absorption coefficient <br> $-\xi$: coefficient controlling the evapotranspiration from the soil layer | $-K_s$, $\theta_{sat}$, $\theta_{res}$, $\psi_{fc}$, $\psi_w$, $\alpha$, $n$ for all surficial and bedrock formations <br> $-C_1$, $C_2$, $C_3$, $A_{root}$, $C_{int}$: parameters of evapotranspiration from soil |
| Saturated flow | $-S_y$, $S_s$, $K_s$: for the groundwater layers | $-S_y$, $S_s$, $K_s$: for the groundwater layers |
| Overland flow | $-\alpha[1/L]$: overland flow coefficient | $-N[L^{1/3}/T]$: Manning number of computational grids |
| River flow | $-$Storage coefficient $[1/T]$ in Linear reservoir routing. <br> $-$Riverbed conductance of the reaches | $-N[L^{1/3}/T]$: Manning number of river reaches <br> $-$Riverbed conductance of the reaches |

Determination of the boundary conditions of the system was the next step of the model set-up process. A no-flow boundary condition was assigned across the lateral extent and the bedrock level of the catchment. The upper boundary condition, however, switches between Neumann (flux boundary condition) and Dirichlet (ponded water level) for the solution of Richards equation in MIKE SHE. In MOBIDIC-MODFLOW, the net precipitation (after the canopy interception) in each time step and the existing ponded water (from previous time step) as well as incoming surface runoff from the neighboring grids determine the upper bound of infiltration. The smallest value between the saturated conductivity and moisture deficit of the gravity reservoir is the lower bound of infiltration [3].

The third step is the initialization of the integrated model. A simulation using integrated surface water-groundwater models requires knowing the water table level and the soil moisture profile in the unsaturated zone of the computational grids to be known. However, such fine resolution quantities of hydrologic variables, even for a small catchment such as Thomas Brook, are seldom, if ever, available and therefore must be estimated. One way to prescribe values to these variables is to perform a free drainage (no precipitation) simulation of the catchment that starts at a full saturated state (water table at the soil surface) until the streamflow at the outlet of the basin matches with a given pre-event baseflow [19,27,28]. Using the parameter values of the MIKE SHE set, the free drainage simulation was performed until the river discharge reached 0.1 m$^3$/s, a value that was measured on 30 July 2004. The resulting water table and soil moisture distributions, the water level in river reaches, and other hydrologic variables were then used in a transient simulation that started on that date and continued until 1 January which is the date from which the two models were executed using an identical initial state. Once MIKE SHE was initialized, the moisture storage of top 1 m of the soil layer was used to assign the initial moisture storage of the gravity and capillary reservoirs of the MOBIDIC-MODFLOW model. For grids with a relatively deep initial water table level, the gravity reservoirs were almost

empty and moisture level of the capillary reservoir is very low. However, in the southern part of the catchment, the grids have groundwater level close to the surface and hence the reservoirs were full. The other initialized variables such as the water table and the water level in river reaches were kept identical in both models. Such initialization strategy ensured a realistic spatial distribution of watershed state variables (soil moisture, water table elevation, water pressure) which in turn guaranteed that the model results could be effectively compared and that any discrepancy between the behavior of the models would not originate from incoherent initial conditions.

The last step in the integrated modeling of the Thomas Brook catchment was executing the simulations. The simulations were carried out for the year 2005, because it was the only year with continuous measurements of outlet discharge and the two monitoring wells (shown in Figure 2). It is important to note that simulations with both models are based on prescribed hydrogeological parameters (see Table 1) and assumed uniform cross-section and roughness values for the river reaches. In other words, neither MIKE SHE nor MOBIDIC-MODFLOW was calibrated against observed streamflow discharges and water tables, as the aim of the study was to compare a simpler (semi) conceptual surface water-groundwater model with a physically based model. Such intermodel comparison has two advantageous: (1) it allows grid-scale investigation of the model behavior for water table levels in both shallow and deep water table grids. This is important as observation of the groundwater levels are often limited to few monitoring wells (two in our case) and calibrating the model parameters based on observed data would not extensively reflect the model performance. (2) Intermodel comparison allows evaluation of the model performance for quantities for which observation does not exist such as groundwater recharge. This allows to investigate how the simplified coupling scheme of MOBIDIC-MODFLOW will affect the accuracy of predicted groundwater recharge considering the response of physically based model as "expected" behavior of the watershed. Nevertheless, we made sure that the uncalibrated models had a reasonably good fit with the observations. Therefore, the comparison between the simulated results of the two models should be able to better show the discrepancies due to differences in model structure, particularly with regard to the conceptualization of the unsaturated zone and its interaction with the saturated zone. Though the two models had similar hydraulic properties for the soil layer, we still needed to calibrate three parameters in MOBIDIC's conceptual scheme i.e., $\gamma$, $\kappa$, and $\xi$ (see Table 2), as well as specific yield in MODFLOW. Specific yield in MODFLOW was calibrated as two models closely predict the water table fluctuations observed at Malcom and Goode monitoring wells. This choice was made as the value of specific yield in shallow water table regions (southern part of the catchment) is much smaller than its value derived from field data. However, the value of specific yield remains constant during the simulation of MOBIDIC-MODFLOW (as opposed to MIKE SHE in which the dynamic behavior of specific yield is captured using water table correction procedure [3]). The $\xi$ parameter controls the magnitude of evapotranspiration from the capillary reservoir, while $\kappa$ and $\gamma$ regulate, respectively, the magnitude of moisture absorption and percolation to the groundwater. IN addition, the conceptual coefficients in MOBIDIC were calibrated based on the simulated response of MIKE SHE in order to have close prediction of moisture balance in the top 1 m soil layer in the two models. Therefore, the coefficients were adjusted since both models have close prediction of evapotranspiration from the soil layer. However, as it will be discussed in Section 5, the models provide a different conceptualization of infiltration and water evaporation, which resulted in differences in total input to the soils. Additionally, differences in conceptualization of the groundwater recharge in the two models resulted in differences in total amount of groundwater recharge for the models. Whereas in MOBIDIC, the calculated percolation in the soil layer is assumed to reach the water table as the groundwater recharge, groundwater recharge, in MIKE SHE, is calculated as the output of the solution of the 1D Richards equation for the whole unsaturated layer.

A key difference between the two models is in time step used in the calculation process. Unlike the use of a single time step in MOBIDIC-MODFLOW (in this case, 1 day), MIKE SHE employs an adaptive time stepping approach in which the actual time step of each module is determined based

on the user specified maximum allowable time step and the numerical stability discussed in [24]. The recommended maximum values of the time step should follow $\max(dt_{river}) \leq \max(dt_{overland}) \leq \max(dt_{UZ}) \leq \max(dt_{SZ})$ in which $\max(dt_i)$ is the maximum allowed time step for the computational module of $i$ [24]. For simulation of the Thomas Brook catchment, the chosen values were 1 min for the river module and 15 min for the other components. Please note that larger time steps for the modules can be selected. However, this requires that the acceptance threshold error of the unsaturated-saturated zone coupling parameters be adjusted. For instance, using 15 min as the maximum time steps for calculation of the unsaturated and saturated flow process allowed to limit the maximum water balance error to 0.001 m. In addition, note that using time steps larger than 15 min resulted in generation of numerical oscillations in simulated water tables.

Regarding the differences in the conceptualization of hydrological processes in the two models, the input parameters (given in Table 2) were kept as consistent as possible. Therefore, the hydrogeological properties of the surficial units and bedrock aquifers (Table 1) of both models were kept identical. The soil water retention characteristics of the formations were defined based on [29] and by assuming that the till layer was the representative surficial formation in the catchment. This is because the soil water retention curves (SWRC) parameters for other surficial formations were not available and since till layer is the dominant surficial deposit of the catchment, fitting parameters of SWRC for all formations were assumed to be similar to those of till as also assumed by [19]. Therefore, the parameters of the model ($\alpha[1/L]$ and $n[-]$) and the residual water content were assumed to be 0.004, 1.7, and 0.001 for all the formations [19]. Additionally, a uniform Manning roughness of 0.1 $m^{1/3}$/s was assigned to both overland and river grid cells as in [19]. Therefore, in this study, the total number of input parameters for MIKE SHE and MOBIDIC-MODFLOW was 44 and 37, respectively.

## 5. Results and Discussion

The simulated streamflow of the two models compared to flows measurements at the catchment's outlet are shown in Figure 8. It can be seen that both models can adequately capture the general trend of observations except for peaks that are not precisely captured by neither of the models. The Nash–Sutcliffe efficiency and percent-bias measures were 0.32 and −0.09 for MIKE SHE and 0.12 and −0.3 for MOBIDIC-MODFLOW. This could be improved with more data on the surface roughness and cross sections of the river networks and by adjusting hydraulic conductivities through model calibration.

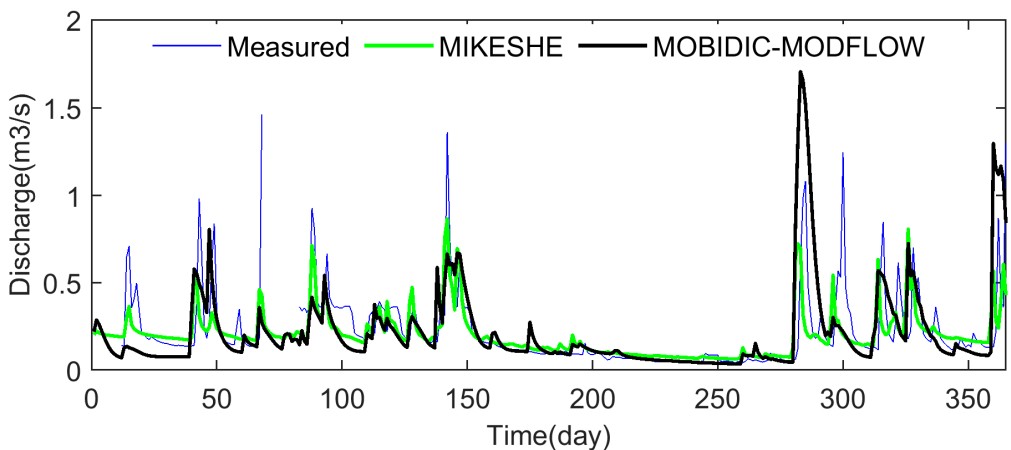

**Figure 8.** Measured (blue) and simulated streamflows at catchment outlet in MIKE SHE (green) and in MOBIDIC-MODFLOW (black), for the year 2005.

In general, MOBIDIC-MODFLOW underestimated the river discharge when compared to MIKE SHE (and to observations) especially in the early days of the simulation, which indicates differences in travel time of the flood wave simulated with the kinematic wave model in MIKE SHE and the linear

reservoir approach in MOBIDIC-MODFLOW. Differences in the spatial discretization of the river reaches made by the two models and the time steps used may also explain in part the differences observed. River reaches in MIKE SHE were discretized into 100 m segments, while in MOBIDIC-MODFLOW, each river reach is considered as a river segment and no discretization of the reaches was made.

In Figure 9, the river-aquifer flow exchanges of the river reaches (reach numbers are shown in Figure 3) are compared. It can be seen that MOBIDIC-MODFLOW underestimates the groundwater baseflow in most of the river reaches which is in part related to the fact that in MIKE SHE, river reaches are placed between the two grids and the model allows flow exchange from both sides of the river (banks). However, in MODFLOW's river package, the flow exchange occurs only in the vertical direction between the river and corresponding groundwater cells. Additionally, differences in river flow routing scheme of the two models affect the calculated water level in the reach and subsequent river-aquifer flow exchange. It is also worth mentioning that the riverbed conductance in MOBIDIC-MODFLOW remains constant during the simulation period, while in MIKE SHE it is a function of water level in river. In reach 16, there is a change in the sign of the flow exchange in MIKE SHE which is due to a very mild slope of reach that results in a higher water level in the river than in the aquifer and subsequently creates negative flows. In MOBIDIC-MODFLOW, however, the simulated flow exchange for this reach is positive for the entire simulation period, which indicates a faster flow propagation in the stream reach. The faster flow propagation in MOBIDIC-MODFLOW is also the reason for the change in sign of flow exchange for reaches 8 and 9 during the late summer especially during the rainy events.

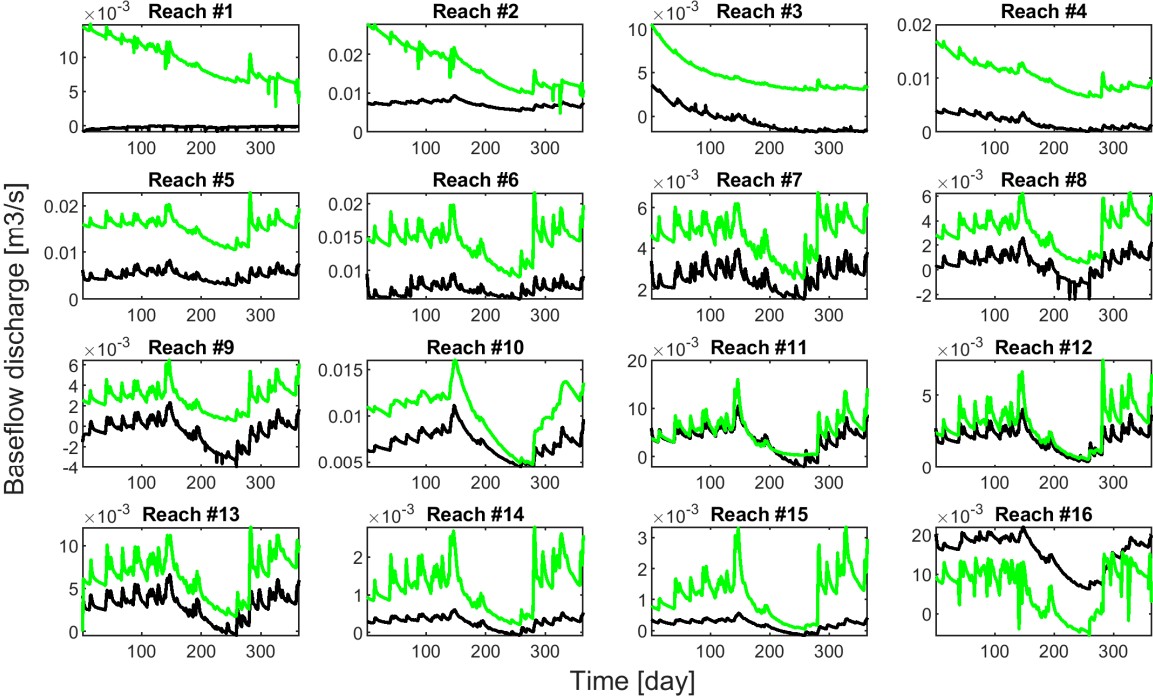

**Figure 9.** Simulated aquifer-river flow exchange of the river reaches using MIKE SHE (green) and MOBIDIC-MODFLOW (black). Note that positive values represent flow from aquifer to river and negative value represents flow from river to aquifer.

The simulated results for groundwater heads at monitoring wells Malcom and Goode (see Figure 2 for locations) are shown in Figure 10. Coefficient of determination ($R^2$) of 0.82 and 0.85 and Root mean square errors of 0.33 and 0.22 m, respectively, for Malcom and Goode indicates a close match between the simulated water table levels of the two models. From Figure 10 it can be seen that the two models have a similar water table decline rate between day 150 and 270, a period that corresponds to summer when groundwater is not much affected by groundwater recharge. This means that the differences

in the calculation of horizontal conductance in MIKE SHE and in MODFLOW have a minor effect on simulation results. Horizontal conductance in MIKE SHE is calculated as the harmonic mean of hydraulic conductivities and the geometric mean of layer thicknesses. In MODFLOW, however, this is derived from the arithmetic mean of saturated thicknesses and the logarithmic mean of hydraulic conductivities. The two models have an identical formulation of vertical conductance, computed as the summation of the inverse of the individual conductance of the two layers.

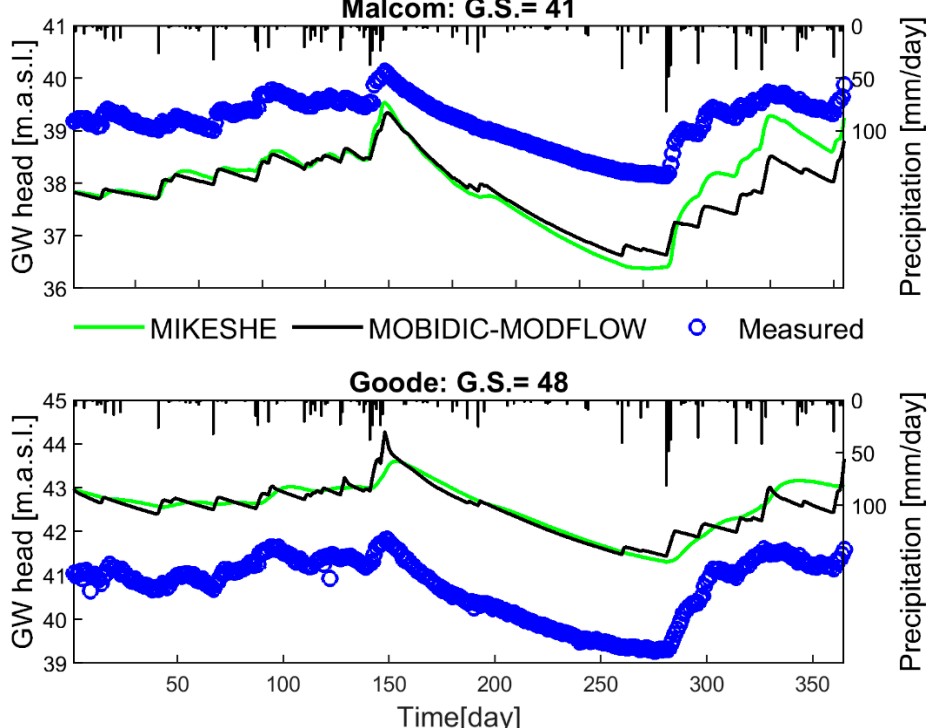

**Figure 10.** Measured (blue) and simulated groundwater head at Malcom and Goode (see Figure 2 for locations) with MIKE SHE (green) and MOBIDIC-MODFLOW (black), for the year 2005. G.S. is the Ground Surface elevation.

The gap between the simulated groundwater heads of both models and the measurements in Figure 10 is related to the initialization process, since the spatial distribution of the water tables associated with the baseflow will not necessarily match the observed groundwater heads, given the fact that the hydrogeological properties of the surficial and bedrock aquifers do not change. This issue was also discussed by others, e.g., [30], as different hydrogeological parameter values of the geological units may give a different spatial distribution of initial groundwater head for the same baseflow discharge. However, as it was stated, our aim was to evaluate the effect of the different structure of the two models on predictions, rather than calibrating their parameter values against observations.

Figure 11 compares the simulated results of groundwater head, infiltration and evapotranspiration rates from the root zone (top one meter) which were obtained from both MIKE SHE and MOBIDIC-MODFLOW during October 2005 (day 274 to 300) at the Malcom observation well. Since the Malcom and Goode grids have relatively deep water tables i.e., 3 and 5 m below the soil surface, the interaction between the modeled soil layer in MOBIDIC (1 m topsoil) and the saturated zone is unidirectional, i.e., from unsaturated to saturated zone. Accordingly, the contribution of the saturated zone to the evapotranspiration (direct water uptake from the water table) is negligible. On day 281, 81.90 mm of rainfall occurred, of which 80.50 mm (98%) infiltrated and replenished the gravity reservoir (see Figure 11b,d). The close match between the predicted infiltration rates of the two models (Figure 11b) confirms the fact that the disparities in predicted water table rises are due to the differences in the formulation of the groundwater recharge of the two models. Note that the potential evapotranspiration

was fulfilled using the evaporation, plus the transpiration from the unsaturated zone in MIKE SHE (Figure 11c). However, in MOBIDIC-MODFLOW, the evaporation from the surface water reservoir (throughfall, plus the existing ponded water from previous time steps) occurs prior to the infiltration process which causes no moisture losses from the capillary reservoir in rainy days (Figure 11c). Such behavior was not observed in MIKE SHE since even in the rainy days the evapotranspiration from the unsaturated zone fulfilled the potential evapotranspiration.

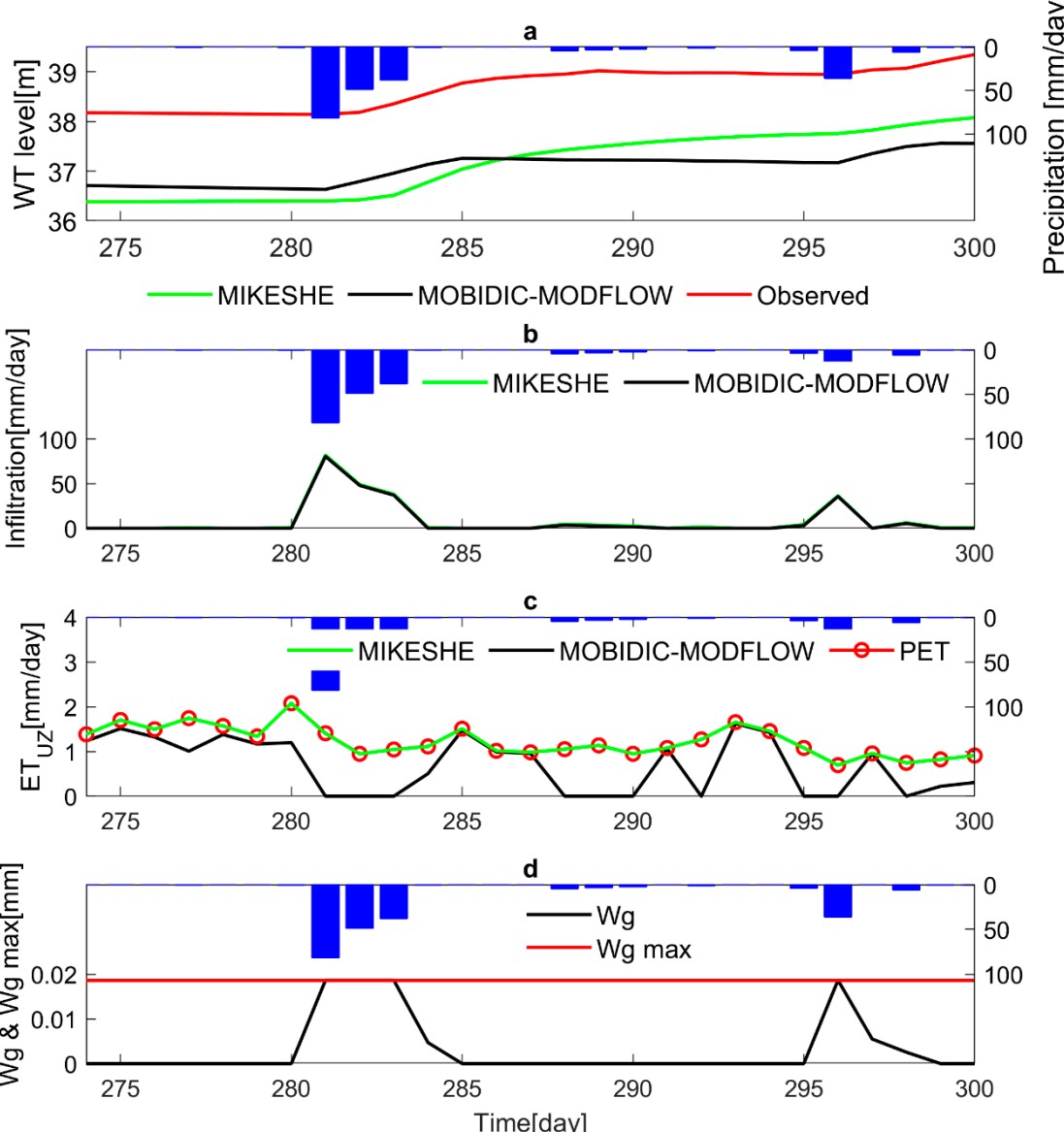

**Figure 11.** (**a**) Observed (red) and simulated groundwater head with MIKE SHE (green) and with MOBIDIC-MODFLOW (black) between days 274 and 300 at Malcom. (**b**) Infiltration rate simulated with MIKE SHE (green) and MOBIDIC-MODFLOW (black). (**c**) Evapotranspiration from the unsaturated zone simulated with MIKE SHE (green) and MOBIDIC-MODFLOW (black) and (**d**) time evolution of gravity reservoir in MOBIDIC-MODFLOW (black) along with the maximum storage capacity of the gravity reservoir (red).

From the infiltrated water, the capillary reservoir drew the moisture required for plant roots, while remaining moisture in the gravity reservoir percolated to the groundwater, causing a water table rise (Figure 11a). The rate of groundwater recharge in MOBIDIC-MODFLOW is controlled by a catchment scale linear coefficient which remained constant during the course of the simulation. This

process continued until day 283, a period in which the gravity reservoir remained full because the deficit caused by percolation to the groundwater and absorption was compensated by subsequent rain. On day 284, there was no rainfall and the moisture level in gravity reservoir decreased and the reservoir became empty on the following day (Figure 11d). This means that no groundwater recharge occurred between days 285 and 295 as precipitation in this period was not sufficient (the infiltrated water transferred to the capillary reservoir to fulfill the unsatisfied transpiration). In MIKE SHE, however, the water table rose during this period (day 284 to 295) and this resulted in the simulated water table position in the two models being different.

In Table 3, simulated and measured groundwater levels at existing groundwater wells in the watershed (see Figure 2 for locations) were compared. Note that the groundwater heads were measured at different times and there is only one value for each well [19]. Wells 1 to 6 are located in the North Mountain formation, wells 7 to 14 in Blomidon, and wells 15 and 16 in the Wolfville formation. Both MIKE SHE and MOBIDIC-MODFLOW closely replicate the measured groundwater heads in the lower part of the catchment (wells 11 to 16) with maximum difference of 1 m. The groundwater levels in the northern part (wells 1 to 5), however, were overestimated by both MIKE SHE and MOBIDIC-MODFLOW. This is due to the presence of abundant vertical fractures in the North Mountain formation which causes the vertical conductivity to be larger than the horizontal conductivity [31]. The formations, however, were assumed to be isotropic for the simulations with MIKE SHE and MOBIDIC-MODFLOW. The overestimation of groundwater heads was also observed when using the CATHY model under the assumption of isotropic aquifer units [19]. The groundwater head at well 6 is significantly underestimated in the two models. This was related to the high rate of river-aquifer flow exchange resulting from assuming there was no low-permeable clogging layer at the streambeds in MIKE SHE and MOBIDIC-MODFLOW.

A key advantage of surface-subsurface hydrological models is their ability to generate spatiotemporal variations of groundwater recharge. Considering the complex heterogeneous structure of the geologic formations of the Thomas Brook and the uncertainties associated with initialization and parametrization of the models, an accurate determination of groundwater recharge is challenging.

The simulated catchment scale variation of groundwater recharge in the two models is shown in Figure 12. The annual groundwater recharge simulated with MIKE SHE and MOBIDIC-MODFLOW are 270.2 and 290 mm, respectively. These values compare favorably with the results of [19], who obtained 315 mm by applying the hydrograph separation method to the observed streamflow; and 349 mm by using CATHY, a physically-based, distributed surface-subsurface hydrological model.

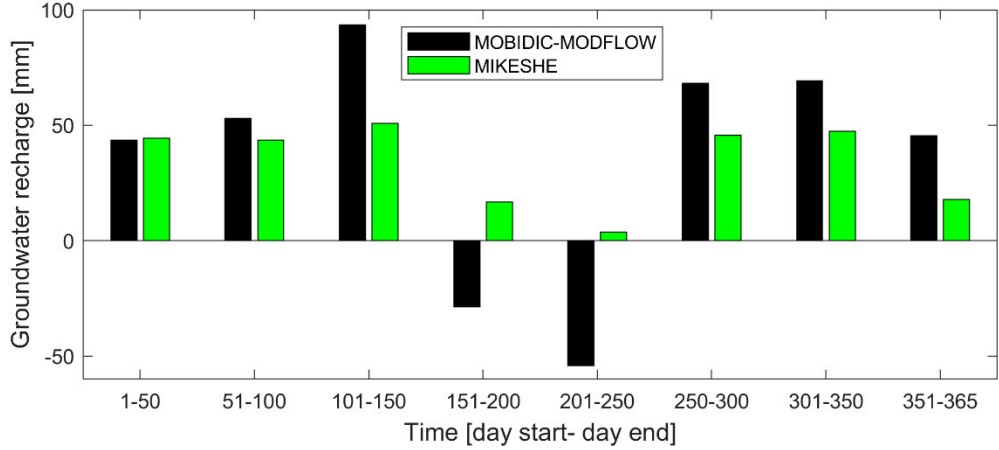

**Figure 12.** Catchment scale behavior of the groundwater recharge simulated by MIKE SHE (green) and MOBIDIC-MODFLOW (black).

**Table 3.** Measured and simulated groundwater heads (in meters) at the groundwater wells shown in Figure 2.

| | North Mountain Formation | | | | | | Blomidon Formation | | | | | | Wolfville Formation | | | |
|---|---|---|---|---|---|---|---|---|---|---|---|---|---|---|---|---|
| | 1 | 2 | 3 | 4 | 5 | 6 | 7 | 8 | 9 | 10 | 11 | 12 | 13 | 14 | 15 | 16 |
| Measured | 201 | 204 | 195 | 185 | 195 | 192 | 57 | 55 | 51 | 52 | 37 | 38 | 39 | 45 | 39 | 30 |
| MIKE SHE | 210.0 | 207.7 | 195.0 | 196.0 | 197.2 | 178.7 | 66.9 | 59.4 | 53.4 | 54.5 | 36.6 | 39.1 | 41.0 | 44.2 | 38.9 | 30.3 |
| MOBIDIC-MODFLOW | 210.7 | 209.0 | 196.2 | 197.5 | 198.2 | 180.4 | 67.4 | 60.0 | 53.8 | 54.5 | 36.3 | 38.9 | 41.4 | 44.5 | 38.7 | 31.0 |

In Figure 12, a similar trend in temporal variations of the groundwater recharge simulated with the two models can be observed. The groundwater recharge increases until day 150 i.e., end of May, decreases during the summer i.e., from day 150 to 250 and increases again during fall and winter i.e., from day 250 to 365. MIKE SHE predicted a smooth increase in groundwater recharge until day 150, while a sharper the rate of increase was obtained with MOBIDIC-MODFLOW. It should be noted that during the winter period, precipitation was dominated by snowfall; hence the magnitude of the groundwater recharge was affected not only by a difference in the conceptualization of the unsaturated flow but also by the snow accumulation/depletion dynamics of the two models.

To exclude the effect of the snowmelt process, the accumulated snowpack on the ground surface of the two models are compared in Figure 13. Both models predict very close snowpack accumulation/depletion especially between day 1 and day 150 (maximum difference of 3 mm) ascertaining that the disparities in groundwater recharge between the two models are mainly due to differences in flow process conceptualization of the unsaturated zone. A comparison of Figures 10 and 12 indicates that the overestimation of the groundwater recharge during the wet period i.e., days 100 to 150 did not affect the water table predictions of MOBIDIC-MODFLOW at Malcom and Goode as the two models simulated very close predictions, even during the spring and fall seasons (see Figure 10).

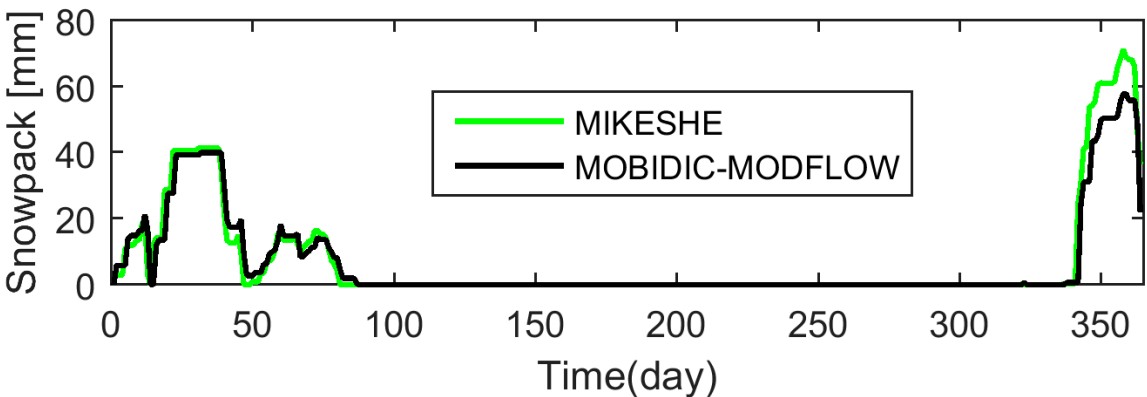

**Figure 13.** Simulated snowpack accumulation/depletion with MIKE SHE (green) and MOBIDIC-MODFLOW (black).

To further investigate this issue, it is important to note that the simulated groundwater heads in both models are affected by (1) total infiltration into the soil profile, (2) evapotranspiration from the unsaturated zone, (3) groundwater recharge, and (4) specific yield

Total infiltration between days 100 and 150 at Malcom for MOBIDIC-MODFLOW and MIKE SHE was 205 and 256 mm, respectively. Since both models show no snowpack during this period, the difference in infiltration is related to the way the evaporation is calculated from the available surface water. In MOBIDIC-MODFLOW, available water evaporates prior to the infiltration to satisfy the unfulfilled evapotranspiration. In MIKE SHE, however, infiltration happens first and evapotranspiration is fulfilled by the soil moisture extracted from the unsaturated zone, unless the water table is at soil surface which results in evaporation of the entire ponded water at the soil surface. Infiltrated water is considered as the upper boundary condition of the solution of the Richards equation in MIKE SHE. The evapotranspiration and groundwater recharge are calculated following the solution of the unsaturated flow module. The calculated groundwater recharge in this period is 128 and 84 mm for MOBIDIC-MODFLOW and MIKE SHE, respectively. The larger value of groundwater recharge in MOBIDIC-MODFLOW, however, did not lead to significant difference between the water tables of the two models since the specific yield in MOBIDIC-MODFLOW was larger than MIKE SHE, 0.07 versus 0.02. As discussed in Section 3.1., specific yield in MIKE SHE is defined as the difference between the water content at saturation and field capacity of the geological unit (see Table 1), and its variations

in shallow water tables are handled using the water table correction procedure in the unsaturated flow module [3]. In MOBIDIC-MODFLOW, however, a constant value for specific yield (determined through the calibration of simulated groundwater heads of MOBIDIC-MODFLOW in comparison to MIKE SHE at Malcom and Goode grids shown in Figure 10) is assumed for the entire catchment.

Note that in MOBIDIC-MODFLOW, if the water table lies below the MOBIDIC's soil layer (top 1 m here), the rate of the groundwater recharge is only controlled by the percolation coefficient and the water table depth does not affect the rate of groundwater recharge. Therefore, increased moisture level in the gravity reservoir due to snowpack melt causes a significant amount of groundwater recharge between days 100 and 150. This is not the case in Richards based models as the unsaturated zone extends to the water table and groundwater recharge is near zero if the water table is far below the soil surface. During the summer period i.e., days 170 to 270, MOBIDIC-MODFLOW predicted negative recharge since most of the grid cells in the central and upper parts of the catchment had empty gravity reservoirs (no recharge to groundwater) and grid cells in the lower parts had high groundwater heads, which resulted in a negative value of groundwater recharge. Therefore, the spatially averaged values of the groundwater recharge obtained by MOBIDIC-MODFLOW during this period were negative. The predicted groundwater recharges in MIKE SHE, however, were not negative but close to zero (see Figure 13). This further reveals the limitations of conceptual schemes in comparison to Richards based models in capturing catchment scale seasonal variations of groundwater recharge.

In Figure 14, the groundwater heads simulated by both models were also compared for six additional grid cells, this time with shallow water table conditions. The initial depth to water table ranges between 20 cm in grid cell 1 to 76 cm in grid cell 5. The comparison of water table response of the two models shows the effects of the simplified unsaturated-saturated formulation approach embedded in MOBIDIC-MODFLOW on the modelling of shallow aquifers, as discussed below.

In Figure 14, one can see that MOBIDIC-MODFLOW, in general, captured the dynamic behavior of the shallow water table, however, the water tables were less responsive than those in MIKE SHE. For example, in tiles 2 and 5, the water table simulated by MIKE SHE quickly rises to the soil surface during mid-winter thaws when snow is melting and during the spring season. Such quick response was not observed in MOBIDIC-MODFLOW. This clearly demonstrates the dynamic behavior of specific yield in shallow water table conditions taken into account in MIKE SHE, which causes much sharper rises of the water table than by using a constant specific yield value as in the case of MOBIDIC-MODFLOW [3]. One must also note the significant difference in water table dynamics between the two models following the rain events occurring in days 281 to 283, which is again related to the way specific yield is handled in MIKE SHE (dynamic values) and MOBIDIC-MODFLOW (static values).

The temporal evolution of the differences in water table depths of MOBIDIC-MODFLOW and MIKE SHE is depicted in Figure 15. Negative values refer to a shallower water table depth simulated in MOBIDIC-MODFLOW in comparison to MIKE SHE, while positive values show the opposite. The differences are very small (a few cm) in most parts of the catchment during the winter period in which, precipitation is mostly snowfall. During the spring MOBIDIC-MODFLOW predicts higher water table (more blue color) compared to MIKE SHE which is because of higher groundwater recharge shown in Figure 12. During the summer, MOBIDIC-MODFLOW underestimates water table level (more red color) as groundwater recharge is smaller compared to MIKE SHE. One must note that the water table predicted by MOBIDIC-MODFLOW along the cuesta (see Figure 1) was quite different than the one predicted by MIKE SHE (differences reach up to 4 m). This is the zone where large variations in hydraulic conductivity (the transition from the North Mountain to the Blomidon formations) and topography (steep slope > 10%) exist. The predicted water tables in MOBIDIC-MODFLOW are shallower (dark blue color) in the vicinity of some river reaches (near the cuesta) and this is due to the fact that the water depth is uniform along a river reach and therefore the exchange flow between river and aquifer (based on differences in head) differs in MIKE SHE, in which river reaches are discretized into computational elements. Additionally, the predicted water tables in MOBIDIC-MODFLOW in the lower parts of the catchment in the three last events are deeper compared to those predicted with

MIKE SHE (red color). This means the specific yield derived from the iterative unsaturated-saturated coupling approach of MIKE SHE is smaller when compared to the MOBIDIC-MODFLOW, which results in shallower water tables, as observed in Figure 14.

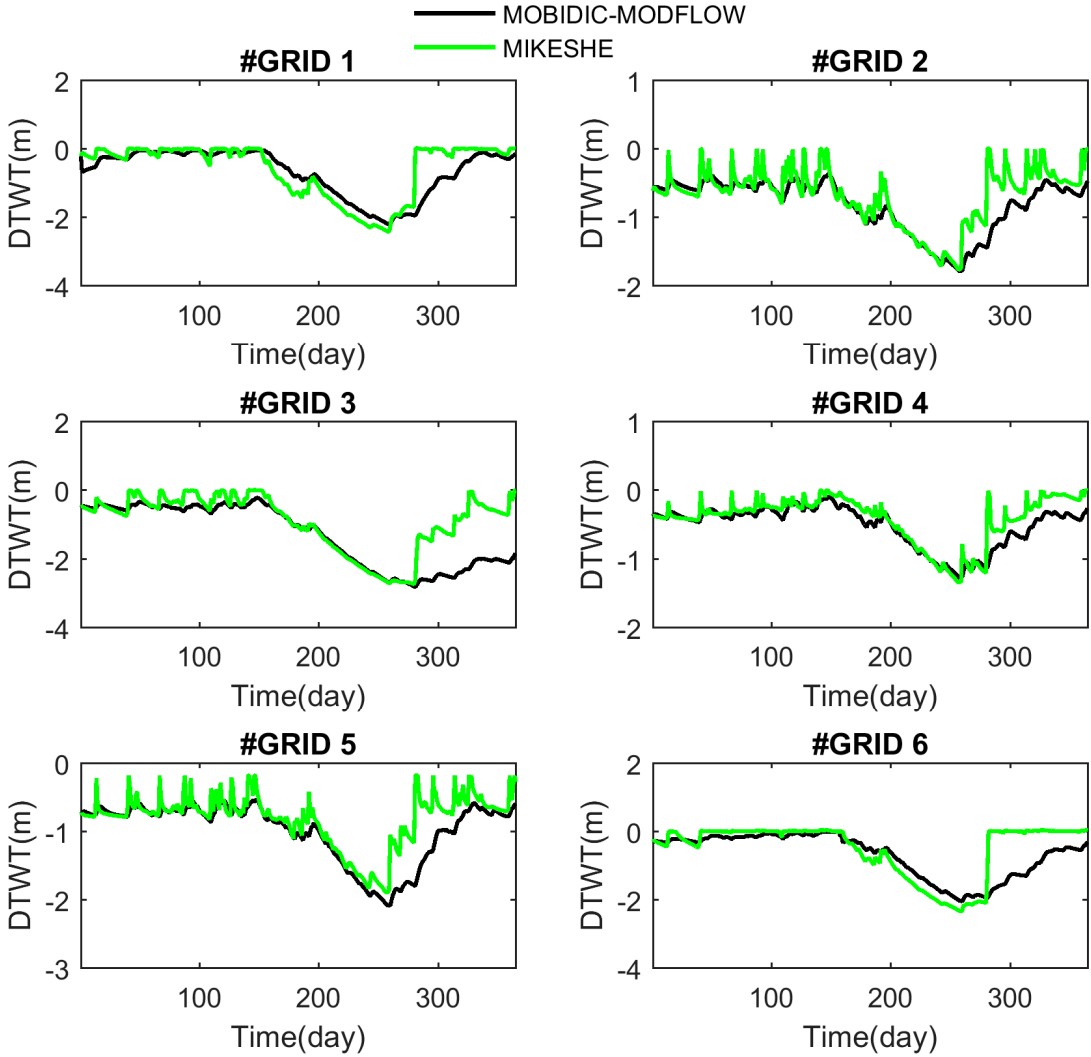

**Figure 14.** Simulated water table fluctuation at six shallow water table grids with MIKE SHE (green) and MOBIDIC-MODFLOW (black). DTWT stands for depth to water table.

Whereas the adaptative time step approach implemented in MIKE SHE can greatly improve its computational efficiency, its execution time is far longer than MOBIDIC-MODFLOW which runs over a uniform daily time step. For example, using a 3.40 GHz Intel core i7 PC, the simulations of Thomas brook catchment with MIKE SHE and MOBIDIC-MODFLOW took about 1 h and 6 min, respectively.

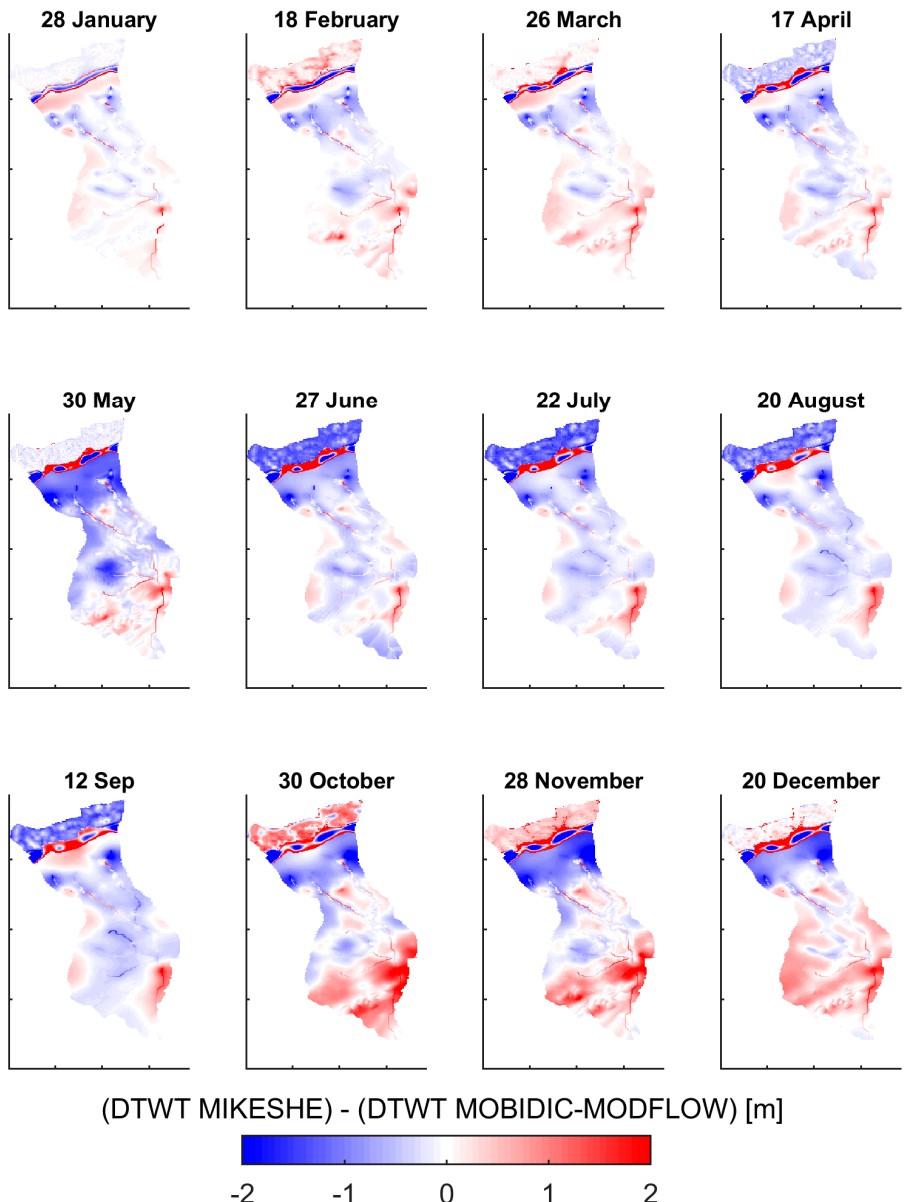

(DTWT MIKESHE) - (DTWT MOBIDIC-MODFLOW) [m]

**Figure 15.** Temporal variation of the differences in depth to water table levels of MOBIDIC-MODFLOW and MIKE SHE. Positive values: deeper water table with MOBIDIC-MODFLOW; negative values: shallower water table depths with MOBIDIC-MODFLOW.

## 6. Conclusions

The interaction between surface water andgroundwater zones in the Thomas Brook catchment was studied using a fully coupled (MIKE SHE) and an externally coupled (MOBIDIC-MODFLOW) surface water-groundwater model. The objective of the study was to assess how the simplified integration approach to groundwater and surface water simulation with MOBIDIC-MODFLOW compares with a physically based model such as MIKE SHE which we took as the reference model for reproducing the spatiotemporal behavior of response variables such as water table levels, streamflow discharges, infiltration rate, and groundwater recharge. The Thomas Brook catchment with complex heterogeneous bedrock and surficial geologic formations and strong variations in elevation values near the North Mountain cuesta allowed for a detailed assessment of the unsaturated-saturated flow interaction approach of MOBIDIC-MODFLOW in both shallow (southern part) and deep (center and western part) water table cases versus the use of a physically based model. Simulations with the two

models were based on identical sets of hydrogeologic parameter values such as saturated hydraulic conductivity and storativity. Additional data required for modelling the unsaturated flow using Richards equation and plant roots distribution were based on available regional [31] and local scale field studies [19]. The coefficients of gravity and capillary reservoirs in MOBIDIC were adjusted as the predicted hydrologic variables with MOBIDIC-MODFLOW closely matched those of MIKE SHE. With regard to the comparison results of the two models, the following conclusions can be drawn:

The two models can adequately reproduce catchment scale water balance components such as infiltration and groundwater recharge. However, the conceptualization of groundwater recharge with a constant spatiotemporal specific yield for the saturated zone in MOBIDIC-MODFLOW resulted in differences in simulated rise and fall of the shallow water table regions of the catchment.

Whereas the monthly variations of the groundwater recharge in the two models follow a similar trend, the calibrated coefficient of the groundwater recharge in MOBIDIC resulted in the depletion of the gravity reservoir during the summer period in the relatively deep water table regions in northern and central parts of the catchment. This reflects the limitations associated with the current conceptualization of groundwater recharge with MOBIDIC for applications requiring contrasting deep and shallow water table regions of the catchment.

The differences in water tables simulated with the two models (Figure 15) show that MOBIDIC-MODFLOW can closely mimic the water table response simulated by MIKE SHE if MOBIDIC is properly parametrized. The differences rarely exceed 1 m in most parts of the catchment. However, the regions close to the river have remarkable differences in water table levels and these are mainly related to the differences in river bed conductance and river-aquifer flow exchange between the two models.

An important advantage of the MOBIDIC-MODFLOW is its computational efficiency. The simulation of Thomas Brook catchment was about 10 times faster with MOBIDIC-MODFLOW than with MIKE SHE. This is of concern in regional scale evaluation of groundwater-surface water interaction studies where available data for the application of physically based models is an issue of concern.

**Author Contributions:** M.B. carried out the simulations with MOBIDIC-MODFLOW and MIKE SHE. He was also involved in drafting the manuscript and interpretation of the simulation results. R.L. and M.N. were substantially involved in all aspects of the work including the design of the simulations, analysis of the model comparisons, and revision of the manuscript. All authors have read and agreed to the published version of the manuscript.

**Funding:** This research was partly funded by a Discovery Grant from the National Science and Engineering Research Council of Canada and by the University of Sherbrooke Water Research Group (USWRG).

**Acknowledgments:** We wish to thank Christine Rivard (Geological Survey of Canada) and Dale Hebb (Agriculture and Agri-food Canada) for providing the required data of the Thomas Brook catchment. We also thank three reviewers for their insightful comments and suggestions that improved the clarity of the paper.

**Conflicts of Interest:** The authors declare no conflict of interest.

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
