# Peer review of "Catchment-Scale Integrated Surface Water-Groundwater Hydrologic Modelling Using Conceptual and Physically Based Models: A Model Comparison Study"

_water, doi:10.3390/w12020363_

Round 1

Reviewer 1 Report

I was asked by the Editor to prepare a short review of your MS. Although you present an interesting methodology for sufrace-groundwater modelling in your manuscript, unfortunately I cannot recommend publication of your manuscript as original research paper in its current shape, for the following reasons:

*       Your paper reads like a project report and needs much more focus on scientific ideas and results..

*       Technical editing is required and no clarity in writing of this manuscript. I had a difficult time in understanding the manuscript.

*       Abstract is quite long. It needs to be condensed.

*       Vertical axis title and information regarding the legends used in figure 11 is required for better understanding

*       In figure 8, there is no good correlation between the observed data and results from analytical solution.

* Check the lines 553 - 583

* Section 2.1 and section 2.2 can be summarize in one paragraph and save 3 pages

* Manuscript suffers from deficiencies of the bibliography; the authors need to provide a more thorough literature review to place their work in context. The paper doesn't really provide any new insights

* Citations are incomplete at many places (figure 1, 5, 6)

* Figures are often difficult to read, e.g. Figure. 1,2,3, 4,5,6,7.

* Latitude and Longitude in figure 1, 2,3 missing

* Others

To conclude, I recommend that this manuscript should be major revised by considering all the above comments cited above, reduce the length of the manuscript and submit it as a Technical Note. Moreover, I am not a native speaker; but I highly recommend a proof reading and and check Plagiarism of your manuscript before any resubmission as well.

Author Response

Dear Reviewer,

Please see the attached pdf file.

Best regards

Reviewer 2 Report

see pdf

Author Response

(The authors gave the same response as above.)

Reviewer 3 Report

This work integrated a distributed hydrologic model (MOBIDIC) with MOFLOW to simulate the coupled surface water and groundwater flow. The coupled model was compared with MIKE SHE in a complex heterogeneous aquifer for predictions such as mass balance and spatially distributed groundwater level. I think this topic is of interest to readers of the Water journal and the study is solid. I especially like the discussion part that the authors investigated the reason for the disagreement between the two modeling approaches. However, the interpretation of these differences might be more straightforward if the authors can use a simple synthetic model for comparison before stepping into a complex case. Overall, I recommend a minor revision of this manuscript.
1. Line 58-65, what is the advantage of this coupled method compared to other externally linked surface water-groundwater model?
2. Line 226 -230, this part needs (one objective of this study) to be mentioned early in the introduction.
3. nFigure 6, two “last time step?” appears in MOBIDIC and MODFLOW parts, are they the same thing?
4. Line 302-304, is there any mesh convergence tests, as the two modeling approaches may have different sensitivities to model resolution.

Author Response

Dear Reviewer 3,

Regards

Round 2

Reviewer 2 Report

The line numbering in the revised manuscript is not correct and inconsistent with the  line numbers provided in the response to the review. This makes it difficult to evaluate the revisions in the revised paper.

However, the quality of the paper and the language has been improved significantly. Most comments appear to have been dealt with. I list below a few remaining comment (first the original comment, then the the response and then my new comment, separated by a line of =)

=============================

Line 443: Figure 9. Explanation of colors is missing. Can simulated water level be added?

Thank you for careful reading of the paper. The description of green (MIKE SHE) and black (MOBIDIC-MODFLOW) was added to the Figure 9 title.

This does not show in the revised paper. Simulated water level has also not been added.

=============================

Lines 394-396: Why have these parameters been defined uniformly and not spatially differentiated?

Thanks for reminding this important point. This is because the soil water retention curves (SWRC) parameters for other surficial formations were not available and since till layer is the dominant surficial deposit of the catchment, fitting parameters of SWRC for all formations were assumed to be similar to those of till. Such assumption was also made by (Gauthier et al., 2009).

It could be considered to add this explanation also to the text of the paper

================================

Line 468: Incorrect figure reference

Thanks for the correction. It as corrected in the revised version of the paper.

Correction not included in revised paper (line 820).

==============================

Line 488: It would be interesting to compare the simulated dynamics also with observed dynamics. Which model captures observed dynamics best?

Thank you for your suggestion. The observed water table level was added to the figure 11 panel a.

Description of the added line is missing in the figure title

==================================

Author Response

Dear reviewer,

Regards
